# Physical Structure Expression for Dense Point Clouds of Magnetic Levitation Image Data

**DOI:** 10.3390/s23052535

**Published:** 2023-02-24

**Authors:** Yuxin Zhang, Lei Zhang, Guochen Shen, Qian Xu

**Affiliations:** 1Department of Traffic Information and Control Engineering, Tongji University, Shanghai 200070, China; 2Shanghai Key Laboratory of Rail Infrastructure Durability and System Safety, Tongji University, Shanghai 200070, China

**Keywords:** magnetic levitation transportation, dense point clouds, incremental structure from motion, multiview stereo vision

## Abstract

The research and development of an intelligent magnetic levitation transportation system has become an important research branch of the current intelligent transportation system (ITS), which can provide technical support for state-of-the-art fields such as intelligent magnetic levitation digital twin. First, we applied unmanned aerial vehicle oblique photography technology to acquire the magnetic levitation track image data and preprocessed them. Then, we extracted the image features and matched them based on the incremental structure from motion (SFM) algorithm, recovered the camera pose parameters of the image data and the 3D scene structure information of key points, and optimized the bundle adjustment to output 3D magnetic levitation sparse point clouds. Then, we applied multiview stereo (MVS) vision technology to estimate the depth map and normal map information. Finally, we extracted the output of the dense point clouds that can precisely express the physical structure of the magnetic levitation track, such as turnout, turning, linear structures, etc. By comparing the dense point clouds model with the traditional building information model, experiments verified that the magnetic levitation image 3D reconstruction system based on the incremental SFM and MVS algorithm has strong robustness and accuracy and can express a variety of physical structures of magnetic levitation track with high accuracy.

## 1. Introduction

The construction of a high-speed magnetic levitation transportation system carries the important mission of developing state-of-the-art (SOTA) key technologies. As a large-scale transportation infrastructure, magnetic levitation transportation faces higher technical requirements in the development of a new generation of information technology. In recent years, the magnetic levitation transportation system followed up the pace of SOTA technologies such as artificial intelligence, smart city, smart manufacturing, digital twin (DT), etc. and became a forward-looking science and technology that can change the traditional transportation industry, which is the key choice facing intelligent magnetic levitation transportation.

Nowadays, magnetic levitation track transportation [1] has increasingly become a promising spot of research focus all over the world. Numerous studies have been performed, and many research scholars have committed to the key technologies [2,3,4] and have maintained continuous growth, such as the guiding method [5] and the track structure [6,7] of the magnetic levitation track. The change and development of the new generation of information technology have made 3D reconstruction technology of image data a new research focus. The 3D reconstruction based on image data primarily includes monocular vision, binocular stereo vision, and multistereo vision. The research on 3D reconstruction of image data mainly includes the structure from motion (SFM) algorithm and the multiview stereo (MVS) vision technology [8,9,10,11]. The SFM algorithm can be divided into many aspects, such as incremental, global, and hybrid, etc. The SFM algorithm takes photos as input information, reduces the camera parameters, and outputs sparse point clouds based on the pixel-matching relationship between the photos. And the incremental SFM algorithm [12,13,14] can recover the camera position and the 3D structural information directly from the overlapping images, which has become a common solution for the current aerial triangulation of unmanned aerial vehicle (UAV) images.

The task of MVS is to reconstruct the real scene closely, usually using a calibrated multiview map, such as getting dense point clouds or voxel representation of the scene. Then it can generate a 3D geometry of the object, and finally, recover the surface of the scene. The research of MVS is mainly divided into three aspects, which are also the three methods of 3D scene representation, that is, voxel-based algorithms, surface slices to cover scene surfaces, and depth map recovery and fusion algorithms. With the rapid development of deep learning, the depth map recovery and fusion algorithms adopted in this paper can make the calculation results more accurate and faster.

In recent years, with the cross-integration of UAV technology and various disciplines, UAV aerial photography technology has developed rapidly. Due to its advantages of high resolution, low cost, and convenient operation [15], UAV photography technology has become an important photogrammetry technology. Current large-scale infrastructure photogrammetry mainly includes UAV, street view cars, mobile phones, and panoramic photography, installed on the rails of vehicles for scanning track photogrammetry methods, of which the most widely used, and the most efficient and convenient, is UAV photography technology. The magnetic levitation image data obtained by oblique photography by UAV, with high definition and large data volume, can provide an effective research basis for magnetic levitation 3D reconstruction. And the 3D reconstruction of magnetic levitation image data can express the physical structure characteristics of magnetic levitation track with high accuracy, providing an effective research basis for cutting-edge technologies such as magnetic levitation digital twin.

The greatest contribution of specific work in this paper is that we realized the effective expression process of 3D reconstruction of magnetic levitation track image data by applying SFM and MVS algorithms. These algorithms make use of the advantages of image data to accurately express the current physical structure of the magnetic levitation track, and the 3D reconstructed magnetic levitation model can provide more important work for further research of DT. Moreover, this paper evaluated the 3D reconstruction system of the magnetic levitation track from the physical structure perspective, and experiments finally prove that the system has strong robustness and accuracy. The remainder of this article is organized as follows: Section 2 reviews the related studies. Section 3 describes the steps and methods of the SFM and MVS algorithms that make magnetic levitation image data to generate dense point clouds. Section 4 analyzes the magnetic levitation scenario. Section 5 expresses and analyzes the experiments and results. Finally, Section 6 and Section 7 summarize all of the contributions of this paper and propose future research directions.

## 2. The Related Work

With the development of computer network science and technology in this century, the 3D reconstruction of image data has become a new technology that can be realized, and it mainly develops into a 3D reconstruction of monocular vision, binocular stereo vision, and multistereo vision. The monocular information is very single, and the depth information cannot be perceived, so the 3D reconstruction of monocular vision requires more complex algorithms and processes, and the reconstruction process will produce errors such as deformation. Binocular stereo vision estimates the depth corresponding to each image and extracts features. Then, it fuses the depth map to obtain dense point clouds and a 3D reconstruction of the scene. The research of 3D multistereo vision reconstruction has gradually developed into SFM and MVS.

As described in Section 1, the incremental SFM algorithm is a common example of an SFM algorithm. Compared with other algorithms, it can add pictures to the 3D reconstruction system one by one, extract feature points, and optimize the 3D point position and camera parameters in realtime, so as to improve the robustness effectively. With the wide application of deep learning, many scholars optimize the SFM algorithm from the perspectives of optimizing the scale-invariant feature transform (SIFT) algorithm [16], the random sample consensus (RANSAC) algorithm [17], feature extraction [18], etc. These algorithms have exploded the field of 3D reconstruction, from autonomous driving, construction, civil engineering, and forestry [19] to smart cities, medicine, and more.

In a retrospective development of MVS in recent years, Vincent L et al. [20]. applied deep learning convolutional neural networks to recover depth map information from image data to restore details of 3D reconstruction. Riegler G et al. [21]. proposed a deep graph fusion framework based on deep learning, which is significantly superior to the traditional signed function fusion method based on mean truncation in noise reduction and suppression of outliers. Yanan Xu et al. [22]. reconstructed smooth homogeneous planar surfaces based on MVS, and this proved the surfaces could be further extended and could provide primitive models for 3D reconstruction. Li et al. [23]. proposed an improved patch-based multiview stereo (PMVS) algorithm for large image sets, and finally showed the availability and practicality of the proposed method.

Among other research methods, many technologies and applications can effectively support 3D reconstruction. For example, Petrie G [24] introduced in detail the operating principles and methods of different oblique photography, including manned and unmanned, the application of UAVs with multiple numbers of cameras, and so on. In the field of 3D reconstruction, there are many scholars [25,26,27] who have actually proved the accuracy and convenience of UAV oblique photography, and that the use of UAVs for mapping and large-scale outdoor 3D reconstruction scenes is feasible and full of potential. UAVs can easily obtain high-definition, large-volume, and all-directional angle scene images, so UAV oblique photography technology is widely used in the field of 3D reconstruction. COLMAP [28,29] is a general-purpose SFM and MVS pipeline with a graphical and command-line interface. It offers a wide range of features for the reconstruction of ordered and unordered image collections. DT was proposed by Professor Grieves [30] in 2002 for product life-cycle management. The NASA definition of DT reflects its elements and purpose. Li et al. [31]. studied the building information modeling (BIM)-based track transportation operation management system, the information sources of DT (including information collected from on-board equipment and trackside equipment), elaborate the data processing system, and the data transmission channel, system linkage, etc. Up to now, DT has become a SOTA research technology in the field of 3D reconstruction and the physical 3D modeling of magnetic levitation track is a basic branch of DT technology, which can provide important research support for smart track DT and smart city DT in the future.

However, the current 3D reconstruction of image data scarcely performs well in macro- and microlevel analyses of the physical structure of large-scale transportation infrastructure, such as magnetic levitation transportation. It is very necessary to explore the 3D reconstruction of large-scale magnetic levitation transportation infrastructure, so as to be aware of the component microstrain, surface corrosion, and macroscopic physical structure track surface deformation in a low-cost way, and can provide timely feedback for emergency repair.

## 3. Magnetic Levitation Image to Generate Dense Point Clouds Method

The incremental SFM algorithm is one of the commonly researched algorithms in the field of multivision 3D reconstruction. It is capable of recovering camera pose parameters of image data and generating sparse point clouds of image data based on the scene geometric relationships between image data feature points. In this paper, we applied feature extraction and matching, triangulation, and bundle adjustment (BA) optimization from the incremental SFM algorithm.

Moreover, we calculated the depth information and normal vector between images from multiple views of image data to recover a 3D stereo model based on the MVS algorithm.

### 3.1. Feature Extraction and Matching

In this paper, two seed images are randomly selected as initialized images from magnetic levitation image data. As shown in Figure 1, these two seed images are random images taken from a UAV at different shooting angles, and the SIFT operator is applied to complete the feature point detection on the seed image data. The SIFT operator is an image local feature extraction algorithm that can find the precise location and principal direction of feature points or key points in different image scale spaces by ensuring rotation invariance and constructing a 128-dimensional SIFT feature vector descriptor to extract features.

Building the Gaussian difference pyramid first.

Number of pyramid groups as expressed in Equation (1):Octave = [log_2_min(*M*,*N*)] − 2, (1)
where *M* and *N* are the number of rows and columns of the original image.

Next, calculate the local maximum or minimum values from the pixel values around the pixel point detected and judge whether it is a key point according to a preset threshold. Then, the main direction of the key point is confirmed, so that the key point feature vector descriptor can be constructed.

Where the scale space radius is expressed in Equation (2):(2)$$r=\frac{3\sigma_{oct}\times \sqrt{2}\times \left(d+1\right)+1}{2}$$
where *σ_oct_* is the intragroup scale of the octave group of key-point.

To match the feature, key point feature vector descriptors are applied to match the seed image feature. We adopt the K nearest neighbor algorithm, which is a commonly used supervised learning algorithm by judging the pixels within the nearest neighbor of a key point belonging to a certain class and by setting the highest threshold so as to connect the feature matching between the key points. In the K nearest neighbor algorithm, the matching problem between objects is avoided by calculating the distance between each object as a nonsimilarity indicator for each object, and this process is based on the Euclidean distance algorithm as Equation (3).
(3)$$d(x,y)=\sqrt{\sum_{k=1}^{n}{\left(x_{k}-y_{k}\right)}^{2}},$$
where *x*, *y* are the coordinates of the pixels.

The final feature point matching between the two seed images is obtained as shown in Figure 2.

Next, we applied the RANSAC algorithm to eliminate the matching error, and the final experiment is shown in Figure 3. The feature matching after applying the RANSAC algorithm can effectively eliminate feature matching error and improve the accuracy of 3D reconstruction.

### 3.2. Triangulation

The triangulation principle of the incremental SFM algorithm focuses on the 3D scene structure information by recovering the camera pose parameters of the image data with key points and gradually adding image data to this system, thus achieving 3D reconstruction of the image data to generate sparse point clouds.

First, calculating and decomposing the essential matrix E by the two seed image data, with their feature matching point pairs, thus can the camera poses of the two seed images be obtained and the 3D points can be generated by triangulation. Next, by adding images to this system continuously and optimizing BA, the 3D reconstruction of the incremental SFM can be achieved.

### 3.3. BA Optimization

BA optimization, as shown in Figure 4, enables nonlinearly optimization of both 3D point locations and camera parameters. After adding a new image, the local BA can only be optimized on the partial images that are most associated with that image. When adding a certain percentage of the model, global BA can be optimized, which can reduce the computational cost.

The Levenberg-Marquardt method is used to solve the minimization reprojection error, as expressed in Equation (4).
(4)$$E(M,X)=\sum_{i=1}^{m}\sum_{j=1}^{n}D(x_{ij},M_{i}X_{j}{)}^{2},$$
where *x_ij_* are the real point coordinates and *M_i_X_j_* are the reconstructed point coordinates.

From Table 1, we can see that the interval between local BA optimization and pose refinement is performed, and each local BA optimization of the SFM system will be iteratively adjusted, so as to change the observations subtly. There are also global BA optimizations with a larger interval, though it is not listed here due to the large amount of data.

### 3.4. Dense Point Clouds Reconstruction

The reconstruction process of the dense point clouds uses the depth map fusion method in the MVS technique [32]. First, a point that can be projected onto multiple images is filtered from the sparse point clouds as a seed point, one of the corresponding image data at that point is selected as the seed image, and the distance from that seed point to the origin of the camera coordinate system of the seed image is recorded as the initial depth of the seed point pixel, and the direction from the seed point to the origin of the camera coordinate system is the initial normal vector. Next, the pixel coordinates around the seed point are projected into space and then gradually expanded to all images to perform nonlinear depth optimization for each seed point, so the depth and normal vector information of the optimized pixel points can be formed to estimate the depth map and the normal map. Finally, according to the depth of each pixel of the depth map, the depth pixel points can be projected into the 3D space using the inverse projection matrix of the camera to obtain 3D dense point clouds.

### 3.5. Method Advantages and Disadvantages

In this paper, we first preprocessed the image data of the magnetic levitation transportation test line acquired by UAV oblique photography, the preprocessed data is performed sequentially to extract the image feature, match the feature, and eliminate matching errors, etc. Next, the 3D scene structure information of camera pose parameters and key points of the image data is recovered and optimized by the BA to output the sparse point clouds of the physical structure of the magnetic levitation track, and these processes are based on an incremental SFM algorithm. Moreover, we applied the MVS algorithm to estimate the depth map and normal map of the magnetic levitation track from the sparse point clouds before and reconstructed the dense point clouds that can express the physical structure of the magnetic levitation track with high accuracy. The whole system is based on COLMAP version 3.7 open-source software to handle the 3D reconstruction algorithm of magnetic levitation image data. The magnetic levitation image to generate the dense point clouds method structure diagram in this paper is shown in Figure 5.

The incremental SFM algorithm has strong system robustness. It can extract feature points from image data information and filter mismatched feature points by the RANSAC algorithm in the feature matching process, so the accuracy and precision of 3D sparse point cloud scenes are high. BA optimization continuously optimizes the sparse point cloud’s 3D structure after adding a new image, local BA only is optimized on the partial images that are most associated with that image. When adding a certain percentage of the model, global BA can be optimized which can reduce the computational cost. However, when the number of images reaches a certain upper limit, each global BA will take much time. The MVS algorithm is suitable for large scenes and massive images and can obtain a number of dense point clouds, however, it much depends on the choice of neighborhood image groups.

## 4. Magnetic Levitation Scenario Analysis

In this paper, we collected image data from the entire 1.5 km magnetic levitation test line of Tongji University Jiading campus in Shanghai, as shown in Figure 6. Our photographic range is above the magnetic levitation line, following the trend of the magnetic levitation line, with a width of 600 m and a length of 2 km of arc-shaped airspace. There were a total of 120 hectares of photographic area photos and the aerial photograph was about 143.3 m from the ground. The specific route map is shown in Figure 7. The collected image data includes the magnetic levitation track grider, bridge decks and piers, turnout, linear line, and turning structure of the magnetic levitation test line, etc. Due to the UAV equipment and the meeting of precision requirements, it has obtained other environmental image data around the magnetic levitation test line.

The turnout structure of the magnetic levitation track, as shown in Figure 8a, starts from the sixth pier at the exit of the magnetic levitation assembly train test line, and it deviates from the magnetic levitation track in about a 5° direction, and the turnout track has six concrete piers as its support. The turning structure of the magnetic levitation track, as shown in Figure 8b, has a total length of 433.44 m, including a variety of linear structures such as gentle curve, flat curve, vertical curve, etc. It is one of the important physical structures of the magnetic levitation traffic test line and the 3D modeling study of it has great research significance for the intelligent magnetic levitation transportation experiment. The linear structure of the magnetic levitation track, shown in Figure 8c, includes a variety of physical structures such as the magnetic levitation track decks and piers, and the magnetic induction coils, which is one of the basic physical structures of the magnetic levitation track.

We obtained magnetic levitation image data by operating the Mini-Scan quadcopter with WIC-61MP camera equipment, adopted the method of oblique photography through the surrounding flight, and shot in five directions of left view, right view, front view, rear view, and downward view. Among them, the four viewing angles of the front and rear, left and right are 45 degrees, and the downward viewing angle is vertically downward. The focal length of the camera is 40 mm fixed focus, and the sensor size is 35.7 × 23.8 mm. Through the above preparations, we acquired a total of 14,590 photos, each with 6000 × 4000 pixels, and each photo occupied about 10 M storage space. In order to reduce the running time cost of this study, and without changing the accuracy of the dense point clouds of the magnetic levitation track, we first preprocessed the image data, manually removed the image data that did not capture the magnetic levitation track (i.e., fully captured the other surrounding environment) from the oblique photography, so that every preprocessed image data can extract the structural features of the magnetic levitation track. In the end, we obtained 3242 images after preprocessing.

This paper applied COLMAP open source software to reconstruct the image data of magnetic levitation track in 3D, applying the main steps of feature extraction, feature matching, sparse point clouds reconstruction, and dense point clouds reconstruction of this software, based on incremental SFM algorithm and MVS technology, and, finally, we reconstructed the dense point clouds that can express the physical structure of magnetic levitation track with high accuracy, such as with magnetic levitation turnout structure, the linear structure, turning structure, etc.

## 5. Experiment and Results

In this study, we preprocessed the magnetic levitation track image data collected by UAV oblique photography and applied COLMAP to successively extract and match features and reconstruct sparse point clouds to finally generate dense point clouds that can express the physical structure characteristics of the magnetic levitation track with high accuracy.

### 5.1. Feature Extraction and Matching

In this paper, we applied preprocessed 3242 magnetic levitation track images to extract features and set and modify initialization parameters such as the maximum number of features, resolution, maximum threshold, etc. Each image was extracted to 8000–12,000 feature points on average. Then we put the feature extracted image data to match their feature, set and modified initialization of the maximum number of matches, maximum ratio, block size, and other parameters, and divided the system feature points into 65 × 65 blocks to compute respectively. Finally, we matched pairs of 407,258 points to complete the data key points research.

### 5.2. Sparse Point Clouds Reconstruction

The image data after feature extraction and matching are reconstructed according to the incremental SFM algorithm to recover the camera pose parameters and 3D scene structure information of key points of the image data and carry out sparse point clouds reconstruction. Finally, this system generated a total of 2,424,880 points to form the sparse point clouds, and the exported point clouds can visualize the overall effect of the 3D reconstruction of the magnetic levitation track image data, as shown in Figure 9. we can see point clouds of physical facilities directly, such as magnetic levitation transportation test lines, residential buildings, factories, etc. and other environmental elements point clouds, such as shrubs, rivers, natural grass playgrounds, bridges, etc.

This section may be divided into subheadings. It should provide a concise and precise description of the experimental results, their interpretation, as well as the experimental conclusions that can be drawn.

### 5.3. Dense Point Clouds Reconstruction

Based on the calculated depth and normal vector information of these pictures’ pixels, we set and modify initialization of max and min pixels number, max depth, normal error, and other parameters, and set the first 20 images with priority as the reference image for seed images are automatically selected during dense reconstruction. Then, we estimated the depth map and normal map as shown in Figure 10.

Finally, the depth pixel points can be projected into the 3D space using the inverse projection matrix of the camera to obtain 3D dense point clouds and matched 149 million dense point clouds as shown in Figure 11.

### 5.4. Dense Point Clouds Effect of the Whole Magnetic Levitation Track

For the dense point clouds of the 3D reconstruction of the magnetic levitation track grider, we performed point clouds data preprocessing manually by data segmentation and other operations to extract dense point clouds that can express the physical structure of the magnetic levitation track grider with high accuracy. The dense point clouds after data preprocessing are mainly divided into three kinds according to the physical structure of the magnetic levitation track grider: turnout structure, turning structure, and linear structure.

The dense point clouds of the turnout structure are shown in Figure 12. We can intuitively know that the turnout grider here is made of cement, which belongs to a typical single-type turnout, and the conversion grider in front of the turnout is made of steel material, which has a larger number of point clouds.

Figure 13 shows the dense point clouds of the turning structure for a magnetic levitation flat curve radius of 400 m.

As shown in Figure 14, the 3D reconstructed dense point clouds can visually and clearly represent the physical information of the linear structure.

### 5.5. Dense Point Clouds Effect of the Details of Magnetic Levitation Track

We made a simulation model of a magnetic levitation track through BIM technology. This simulation model is based on the basic data of the magnetic levitation test line of Tongji University Jiading construction, such as track width, straight line length, track grider length, circular curve radius, and transition curve parameters, etc. Then, we mapped the basic model according to the image data and we relied on the material of the track grider.

For cement material and steel material, so as to build a BIM model that can restore the magnetic levitation track on the surface, the whole process is applied with professional software, such as AutoCAD, 3DSMAX, Context Capture, etc. This BIM model can be compared with our dense point clouds model. Figure 15 shows the local detail comparison of the magnetic levitation turnout structure.

As shown in Figure 14, it can be intuitively expressed that the difference between the traditional BIM model constructed by static drawings and the dense point clouds model reconstructed by applying image data to 3D. First of all, the BIM model reflects the data of the track surface at the beginning of engineering construction, however, the actual construction will have systematic errors. These data cannot be presented from BIM models, while the dense point clouds model is formed by image data, which is scanning the current track entity and can show physical differences. This magnetic levitation track test line has been in operation for more than two decades, which could not be estimated by the BIM model due to the changes caused by the track vibration test, physical friction, sensor aging, and acid rain corrosion, though the dense point clouds model has the current collected data, which can provide more economical and convenient feasible measures for aging sensors maintenance, and can also provide better real data for magnetic levitation DT applications. Therefore, the dense point clouds model has more objective research value than the BIM model from the perspectives of economy, application value, and future development prospects.

In order to verify the 3D reconstruction effect of the dense point clouds model, we compared the track width, track length, building height, turnout length, etc., as shown in Table 2 with the basic data of the BIM model, which is hardly affected by time.

Table 2 shows that compared with the basic data of the BIM model, the accuracy rate of the dense point clouds model is higher than 97%. The error is controlled in the millimeter range, and the overall effect of the 3D reconstruction is very good, so it can be proved that the 3D reconstruction model of the image data based on the SFM and MVS algorithms has good robustness.

## 6. Discussion

In the field of large-scale transportation infrastructure and magnetic levitation transportation, we applied SFM and MVS algorithms to realize the 3D reconstruction process of image data collected by UAV oblique photography technology, which integrates engineering, computer science, applied physics, and other disciplines. In this paper, it is an integrated innovation between the technology and application levels. In addition, we overcame the application limitations of large data on the algorithm and can ensure the accuracy of the model and the robustness of the system. However, we also faced some difficulties with long calculation times. There was a dilemma with a long global BA optimization time and we hope that could be solved in follow-up experiments.

In general, our experiment can expand new fields for current ITS technology, break the traditional restrictions on the maintenance and overhaul of magnetic levitation large-scale traffic, provide basic research for magnetic levitation digital twin technology, and provide application value for smart cities.

## 7. Conclusions

In this paper, we extracted magnetic levitation image feature information and feature matching based on an incremental SFM algorithm, recovered camera pose parameters of image data and 3D scene structure information of key points, and optimized BA to generate sparse point clouds data, and then calculated and optimized depth map and normal map by MVS algorithm, and output a magnetic levitation track that can accurately express its physical structure information such as turnout, turning and linear structures. Finally, we compared the basic data of the BIM model with the modeling data of the dense point clouds model, and proved that the accuracy of the dense point clouds model is higher than 97%. Therefore, the model we reconstructed has a better expression effect, the system has strong robustness, and it can better show the real situation of the current magnetic levitation transportation infrastructure.

We reconstructed high-precision sparse point clouds of magnetic levitation track based on incremental SFM algorithm with COLMAP version 3.7 software;We calculated dense point clouds based on the MVS algorithm that is able to express the physical structure of the magnetic levitation track, such as turnout, turning, and linear structures, with high accuracy;The magnetic levitation dense point clouds model in this paper is easy to integrate with other 3D reconstruction models for further research, such as the laser point clouds model.

The research of intelligent magnetic levitation image 3D reconstruction can provide a significant scientific basis for the further development of ITS industry technologies such as magnetic levitation digital twin, smart city, etc.

## Figures and Tables

**Figure 1 sensors-23-02535-f001:**
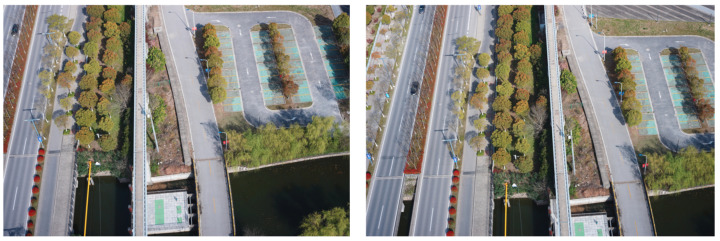
Two randomly selected magnetic levitation image data as seed images.

**Figure 2 sensors-23-02535-f002:**
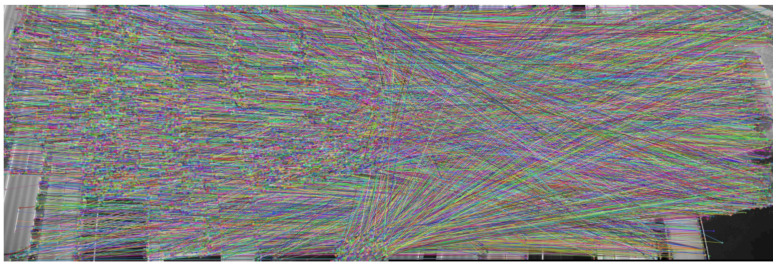
Seed image feature point matching.

**Figure 3 sensors-23-02535-f003:**
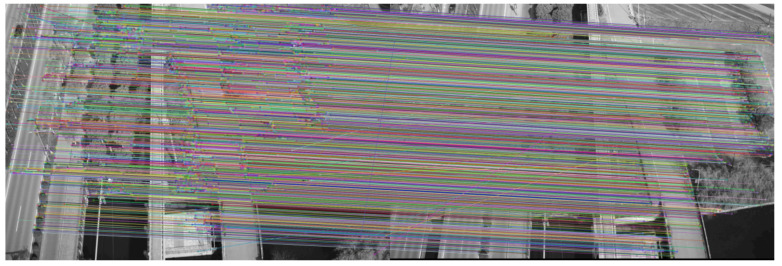
RANSAC algorithm to eliminate seed image feature matching error.

**Figure 4 sensors-23-02535-f004:**
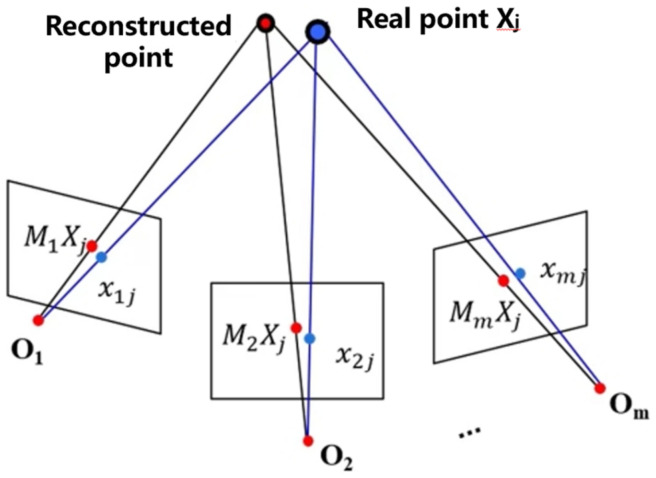
BA optimization decomposition.

**Figure 5 sensors-23-02535-f005:**
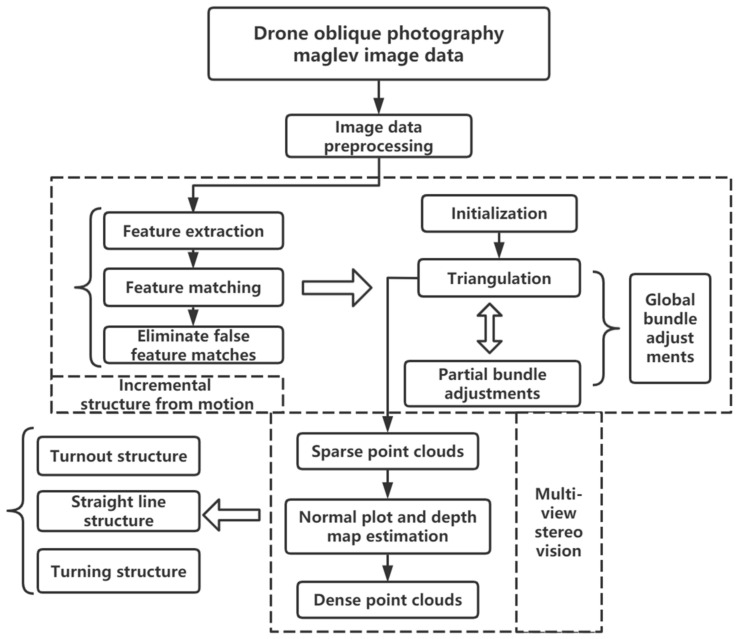
Structure of magnetic levitation image to generate dense point clouds method.

**Figure 6 sensors-23-02535-f006:**
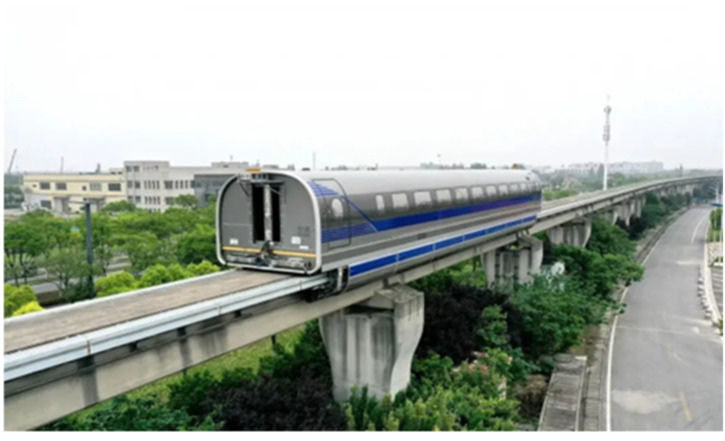
Magnetic levitation transportation test line at Tongji University’s Jiading campus.

**Figure 7 sensors-23-02535-f007:**
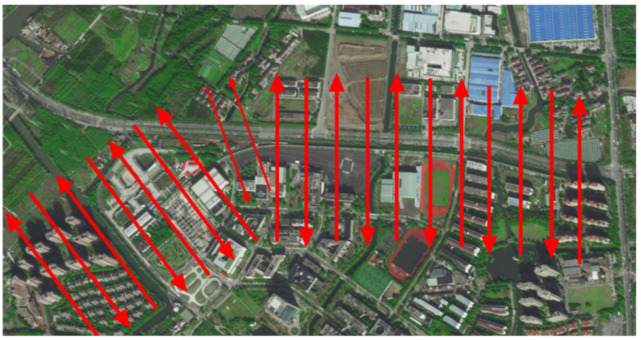
Oblique photography route plan of the magnetic levitation test line.

**Figure 8 sensors-23-02535-f008:**
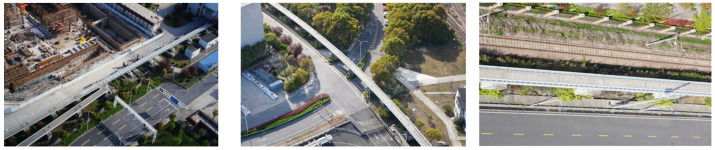
UAV images of the physical structure of the magnetic levitation track grider: (**a**) Turnout structure; (**b**) Turning structure; (**c**) Linear structure.

**Figure 9 sensors-23-02535-f009:**
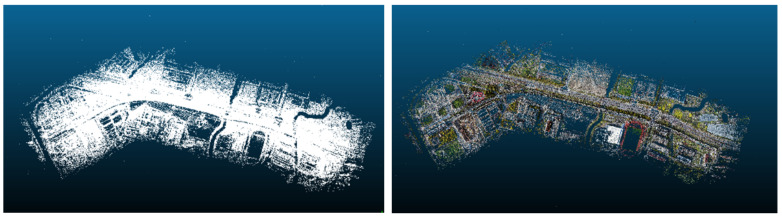
Overall effect of sparse point clouds of magnetic levitation track: (**a**) The color of points are empty; (**b**) The color of points with RGB.

**Figure 10 sensors-23-02535-f010:**
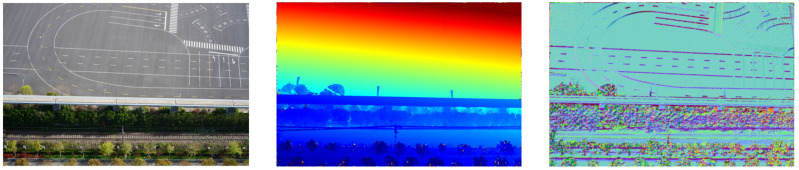
The original image and depth image and normal image of one picture.

**Figure 11 sensors-23-02535-f011:**
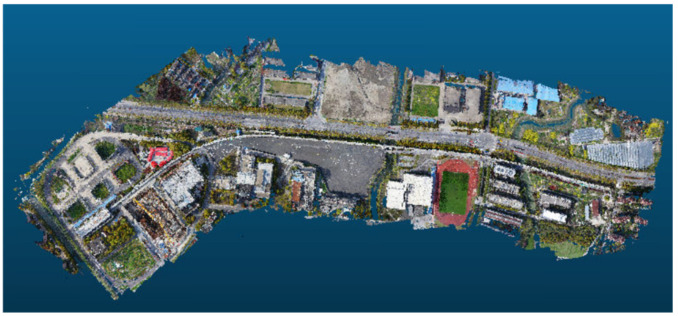
Overall effect of dense point clouds of magnetic levitation track.

**Figure 12 sensors-23-02535-f012:**
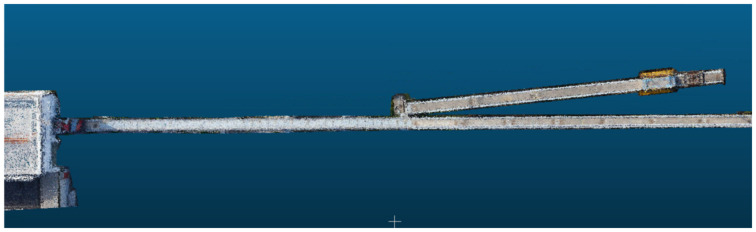
Illustration of the dense point clouds of the magnetic floating turnout structure.

**Figure 13 sensors-23-02535-f013:**
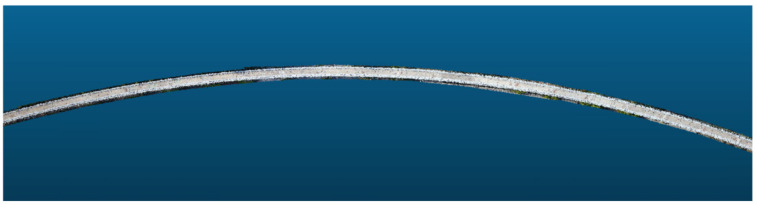
Illustration of the dense point clouds of the magnetic floating turning structure.

**Figure 14 sensors-23-02535-f014:**
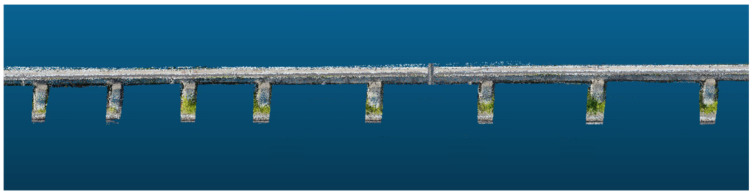
Illustration of the dense point clouds of the magnetic floating linear structure.

**Figure 15 sensors-23-02535-f015:**
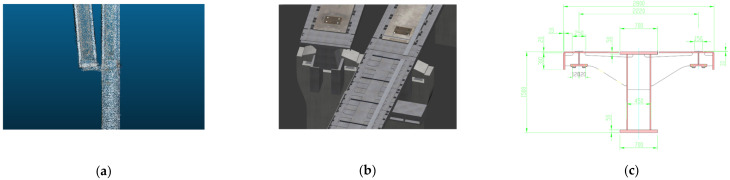
Local details of turnout structure: (**a**) dense point clouds model; (**b**) BIM model; (**c**) standard cross-sectional turnout structure diagram.

**Table 1 sensors-23-02535-t001:** BA optimization and Pose refinement data process.

Operations	Residuals	Parameters	Iterations	Initial Cost	Final Cost	Changed Observations
Local BA	58,332	10,490	4	0.750954	0.736611	0.007336
Pose refinement	5842	6	10	0.744848	0.74296	-
Local BA	50,238	6251	26	0.743882	0.717799	0.028914
Local BA	47,640	6170	3	0.870183	0.861505	0.007159
Pose refinement	9134	6	8	0.69557	0.693146	-
Local BA	62,046	8531	26	0.730047	0.702608	0.044106
Local BA	60,616	8108	3	0.778276	0.766538	0.008588
Pose refinement	5566	6	11	0.778609	0.769827	-
Local BA	60,448	10,019	26	0.822359	0.774519	0.053219
Local BA	58,566	9383	4	0.781542	0.76135	0.008750

**Table 2 sensors-23-02535-t002:** Comparison of data accuracy between the BIM model and dense point clouds model.

Location of Measure	Length (m)	Model	Accuracy (%)
Track width	18.576	BIM	Default
	18.592	Dense point clouds	99.9
	18.434	Dense point clouds	99.2
Track length	2.800	BIM	Default
	2.799	Dense point clouds	100.0
	2.738	Dense point clouds	97.8
	18.086	BIM	Default
Workshop	17.893	Dense point clouds	98.9
	17.863	Dense point clouds	98.8
Turn length	80.689	BIM	Default
	79.853	Dense point clouds	99.0
	79.674	Dense point clouds	98.7

## Data Availability

Due to privacy restrictions, we cannot provide information on the use of data.

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
