# Peer review of "Physical Structure Expression for Dense Point Clouds of Magnetic Levitation Image Data"

_sensors, 2023, doi:10.3390/s23052535_

Round 1

Reviewer 1 Report

In this paper, the authors adopted incremental SFM algorithm and MVS technology, using COLMAP software and carrying out UAV photography, image data preprocessing, feature extraction and matching, triangulation, bundle adjustment optimization, dense point cloud reconstruction, etc., to realize the physical structure 3D reconstruction of maglev line. The study is of positive significance for the digital twinning of maglev track line.

However, there are the following points that need further clarification:

1) The incremental SFM algorithm and MVS technology adopted in this paper are mature technologies in the field of computer vision image processing. This paper adopts this method to obtain the physical structure of maglev test line, which belongs to the conventional technology application. Thus come the questions: What are the innovation points reflected in the research? What are the contributions of this research to the reconstruction of physical structures? What problems does the research solve in the related research field?

2) As described in the abstract of the paper (Line 20) : The experiment verifies that the 3D structure reconstruction based on incremental SFM and MVS technology has strong robustness and accuracy, but there is no relevant data and description in the paper to support it.

3) The paper carries out data point cloud reconstruction for the physical structure of maglev line, but what is the digital twin relationship between the physical structure reconstruction of line track and the function and performance of maglev system? Which chapters in the paper show this digital twinning?

4) Line235-237, Line274-275 seems to be a suggestion for paper writing, rather than the research content or conclusions of the paper.

5) "it much depends on..." in the last sentence of Line167. "much" is not grammatical.

Author Response

Thank you very much for Prof‘s’ review, which provided me with a very valuable experience in my research work.

We compared the data with the model presented in this article according to the BIM model, and improved the relevant data and description in the paper to support in the Experiment and Results chapter, and also adds The Related Work chapter, introduces the algorithms and software applied in this article more in detail, highlights the advantages and disadvantages of the model in the Discussion and Conclusions sections, and corrects the details you remind the error.

Reviewer 2 Report

The manuscript seems feasible and interesting in the area of transportation. The authors need to revise the manuscript to publish the article in Sensor, MDPI.

1. Improve the overall English grammar of the manuscript.

2. Authors must include an Introduction section where background, related work, motivation, and contribution.

3. Authors need to add more sections related to the manuscript i.e methodology, image analysis, etc.

4. Figure 1 both image seems identical (justify).

5. Comparative analysis of the work must be done to validate your work.

6. More data on BA optimization is needed to be added in section 2.3.

7. In section 4 figures must have some labels inside for better understanding for the readers.

8. Authors are advised to include future directions of the study.

9. You must highlight the novelty of the work in the respective section.

10. Lastly, authors must include state-of-the-art references, especially from the last three years.

The above mention suggestions are required for the betterment of the manuscript. 

Author Response

I am very grateful for Prof's review comments, which have helped me very meaningfully in my research career.

According to your review suggestions, I focused on adding the Introduction, The Related Work chapter, and re-revised and added many relevant references in the past three years; In the most important chapter of Experiments and Results, I have added the data comparison between the model in this paper and the traditional BIM model, and proved the robustness of the model; In addition, I also corrected some gramma errors, and highlighted the advantages and disadvantages of the model and future research directions in the Discussion section; The two pictures in Figure 1 are photos taken from the same location but different angles by UAV, and I changed the Figure 1 description; BA Optimization section I updated the data Table 1, too.

Thanks again to the professor for your review suggestions, I am extremely grateful.

Reviewer 3 Report

The paper deals with the photogrammetric use of UAV imagery for survey magnetic levitation infrastructures. Photogrammetry is then the principal point of view considered. Unfortunately, from this aspect the paper is very weak since the processing and the results were faced in very simple and elementary mode.

Introduction does not explain which is the state of the art of magnetic levitation infrastructures survey. How are they survey now (with which technique)? What is missing? What is the advantage of using UAV imagery? Is there any possible limitation? What about accuracy needed for this type of elements?

Methodology section is quite good even if it is possible to find this type of explanation on SfM and MVS in many papers. All is wrote quite correctly but in a superficial ways.

About the UAV missions, it is not clear how they were planned. Did you perform only longitudinal acquisition or cross strips too? You used five oblique angles: which are theirs values? Did you use any GCPs (Ground Control Points)? How did you manage camera calibration?

Results are the most weak part. The paper shows some figures obtained by the photogrammetric process and this is very trivial (showing the sparse point cloud with or without RGB information is not particularly scientific). What about quality check? What about the accuracy of you processing? You segmented the dense point cloud but in which way: manually or automatically?

Finally, the paper contains many typos (uppercases missing) and it seems that it was not read carefully because several sentences are very repetitive. Another example is the first sentence of Section 5 where there still is a sentence of the template file.

Author Response

Thank you very much for professor's review, which is very important to my research life.

According to your review comments, I have rewritten the Introduction, The Related Work chapter, and described the algorithms applied in this article and the application status; I focused on updating the chapter of Experiments and Results, and compared the dense point clouds model with the BIM model, which proved the accuracy and robustness of dense point clouds model in this paper; In the Discussion chapter, the research significance, advantages and disadvantages and prospects of this paper are introduced; Data segmentation is done manually, and these are also marked in the article; I also corrected some grammatical errors in the article; The oblique photography technology of UAV is an application technology for collecting image data in this paper, and this article focuses on the application processing part of image data, so only some parameters of UAV are introduced.

Thank you again for prof's valuable review, I couldn't be more grateful.

Round 2

Reviewer 2 Report

I have reviewed the manuscript again and found satisfactory. The authors did great revision and performed all corrections as suggested. Now the manuscript is recommended for publication in mdpi.

Author Response

Thank you very much for your review comments, it helped me a lot, and I wish you a happy life

Reviewer 3 Report

Dear authors,

I appreciated the effort to improve your contribution but many issues are still unsolved. I will try to explain which points are still need a deeper analysis.

First issue:

You have extended the introduction adding a state of the art section but it only refers to photogrammetric and computer vision topics. As I already wrote, it should be more interesting to analyse what it is currently done for magnetic levitation infrastructures survey. I repeat again my questions: how are they currently surveyed (with which technique)? What about accuracy needed for this type of elements?

Explaining this aspect allows to understand as a UAV survey could be interesting in ITS. I am not a transportation expert but I’m a photogrammetrist and it is not clear for me this aspect. This explanation could also be interesting to some readers to understand the adding value of UAV in relation to traditional measurements.

Second issue:

As your paper concerns a photogrammetry project you must write something more about it. How did you plan the flights? Only longitudinal or also cross strips? How many strips? Did you use any GCPs? What about camera calibration?

This aspect is particularly important in case a readers would replicate your experience. Surveying linear structures are particularly difficult especially when they are many kilometres long. A robust geometry is mandatory in this case. A detailed description of your experience are then very important.

Third issue:

For data quality control, you compared dense point clouds and BIM. Also in this case many questions still remain without any answer. How did you perform this comparison? Which is accuracy of you BIM (which was the original data use to construct the BIM model)?

As reported before, linear structures are challenging for surveyors and data control is important to understand the quality of the acquired data. In this case, the comparison are usually performed using ground truth that is theoretically better than the tested data. I’m not sure that a BIM model could be an useful ground truth.

For these reasons I think that your contribution must be further improved.

Author Response

It is my pleasure to have an expert in the field of photogrammetry guide my academic papers. But allow me to present my related research, on the oblique photography part.

We have added the specific steps of the experimental part of oblique photography to obtain image data, as well as the relevant parameters in detail, according to your review comments, please refer to the magnetic levitation scene analysis chapter for details.

We also added the development of oblique photography-related techniques and their importance, respectively in sections 1 and 2. And the relevant reference list is also enriched.

Maybe my research on the oblique photography part still doesn't satisfy you, sorry, because we just used UAV oblique photography as a method of obtaining image data, but we still give a full explanation of the relevant steps, hoping to provide a small reference for other scholars.

In addition, the accuracy of the BIM model you mentioned, the BIM model we did can provide a data comparison basis for future research, and its error is within the allowable range of the BIM model, and the part we chosen for data comparison in this article is the part of the maglev line that is not affected by loss, time and other factors, such as track length, so it can be used as a scientific support for model accuracy testing.

Thank you again for your valuable review comments, which helped me a lot.  Please see the attachment
